# The effect of gamma irradiation on the stability of vitamin D in select finfish species

Jessica S. Brown[1]*, Patricia R. Calvo[2], Pujita Julakanti[1], Fatima Mohiuddin[1], Abdullah Basir Khan[1], Kemly Julien[1], Varun Natarajan[1], Leonardo B. Maya[3], Kaylyn A. Keith[3], Anthony P. DeCaprio[3], Ryan Hollingsworth[4], Frank Benso[4], Robert P. Smith[5,6]

1 Department of Chemistry and Physics, Nova Southeastern University, Fort Lauderdale, Florida, United States of America, 2 Department of Chemistry, Kansas State University, Manhattan, Kansas, United States of America, 3 Forensic & Analytical Toxicology Facility, Global and Forensic Justice Center, Florida International University, Miami, Florida, United States of America, 4 Gateway America, Gulfport, Mississippi, United States of America, 5 Cell Therapy Institute, Kiran Patel College of Allopathic Medicine, Nova Southeastern University, Fort Lauderdale, Florida, United States of America, 6 Department of Medical Education, Kiran Patel College of Allopathic Medicine, Nova Southeastern University, Fort Lauderdale, Florida, United States of America

* jbrown3@nova.edu

## Abstract

Finfish are an excellent source of dietary vitamin D and additional nutrients. Accordingly, their global consumption has increased. To meet this demand, finfish are now imported across large geographic distances. This poses several challenges, including the potential contamination of finfish with pathogenic bacteria, which can cause food-borne illnesses. Gamma irradiation is a potential solution with a history of reducing bacterial density in foods without compromising nutrients or taste. While some previous work has suggested that vitamin D is stable during irradiation, others have suggested that irradiation may reduce vitamin D. Importantly, these studies have not been completed in finfish tissue of commercialized species, which may offer cross-protection against any reduction in vitamin D. However, this has yet to be evaluated. In this study, three filets each of salmon and trout were dissected into multiple sections and each section was exposed to a different dose of gamma irradiation under both chilled (4°C) or frozen condition (−17°C). Here, we show that the stability of vitamin D during irradiation depends on finfish species and temperature in a dose-dependent manner. Specifically, we found that there was no significant change in vitamin D when trout was irradiated in the chilled or frozen state. Conversely, salmon showed a significant decrease in vitamin D when radiation doses exceeded 0.5 kGy in the chilled state and 2 kGy in the frozen state. Overall, our results indicate that irradiation of finfish may not reduce vitamin D concentration when applied at dosages of 0.5 kGy and 2 kGy or less in chilled and frozen conditions, respectively. Consequently, irradiation may represent a mechanism to increase the safety of consuming finfish while not impacting an important source of dietary vitamin D.

**Data availability statement:** All raw data associated with the manuscript can be found in figshare (https://doi.org/10.6084/m9.figshare.30128818.v1).

**Funding:** Robert P Smith received funding from the Seafood Industry Research Fund.

**Competing interests:** Robert Smith is an author on a petition to the US FDA that seeks the approval of irradiation for finfish and flatfish.

## Introduction

Finfish are regarded as an excellent source of many essential nutrients and vitamins, including vitamin D. This is being increasingly recognized by the population as *per capita* consumption rate of finfish and their products has increased. For example, over the past 30 years, a marked increase in finfish consumption has been noted in the United States. In 2021, Americans consumed an average of 20.5 pounds of finfish and shellfish, representing the highest average consumption weight of seafood reported to date [1]. It is expected that to satisfy the increased demand for finfish, the amount of imported finfish sources from geographically distant regions will increase [2].

Importing finfish from geographically distinct regions poses several challenges, including an increased chance for spoilage and contamination with pathogens [3,4]. Among the many bacteria that can contaminate finfish, *Vibrio* sp., which contaminates finfish in their natural habitat or during the handling and processing phases, is of increasing concern [3,5]. Indeed, the number of individuals infected with *Vibrio* sp. has generally increased [6]. Infections from these pathogens can be fatal; approximately 50% of immunocompromised patients with *V. vulnificus* septicemia die [7]. Widespread outbreaks of *V. parahaemolyticus* have been associated with the consumption of seafood [5]. Strikingly, 18% of foodborne illnesses in the United States have been attributed to the consumption of finfish contaminated with pathogens [8]. Overall, there is a need to identify, develop, and implement strategies that reduce the incidence of contamination of finfish with *Vibrio* sp. and additional pathogens that cause foodborne illness.

One strategy that has a proven track record of reducing the density of bacteria [9–12], including *Vibrio* sp. [13], *Listeria* sp. [13], and *Clostridium* sp. [14,15], in foods while maintaining food quality is gamma irradiation. In the irradiation process, ionizing energy is applied to the food, which kills bacteria by causing DNA damage [16,17]. The dose of irradiation required to eliminate bacteria is temperature dependent as multiple studies have shown that higher doses of gamma irradiation are required to inactivate bacteria at subfreezing temperatures [18]. Irradiation can also extend the shelf life of foods [18,19] while maintaining key nutrients and organoleptic properties [20–22]. Multiple countries have implemented the irradiation of finfish to reduce the density of pathogenic bacteria and to increase food safety [17]. Furthermore, irradiating chicken, beef, lettuce, and crustaceans is already commonplace in the United States [23].

While irradiation represents an established strategy that could reduce the incidence of pathogens found in finfish, the objective of this study was not to evaluate the microbial efficacy of irradiation, but rather to assess how radiation dose and temperature affect the stability of vitamin D in finfish tissue of commercial species. This is critical to address, as finfish represents one of the only sources of food-acquired vitamin D. As many people are regarded as vitamin D deficient [24], any loss of dietary vitamin D during irradiation could have a negative health outcome. Previous work performed on whole extracts of finfish [10], vitamin D resuspended in organic

solvents [10], and chicken [25] has yielded somewhat conflicting results. On the one hand, some studies show that irradiation does not impact the concentration of vitamin D [26]. On the other, some previous work has shown that irradiation can reduce the concentration of vitamin D [10,27]. However, to date, not a single study has investigated the stability of vitamin D in commercially important, fresh and frozen finfish species. This is important to consider as the composition of the chemical matrix where vitamin D is found may react differently to irradiation and may provide protection against any loss. To ensure practical relevance, we chose to irradiate finfish samples under chilled and frozen conditions that reflect how these products are commonly stored and sold commercially. To address this knowledge gap, we measured the effect of irradiation on vitamin D in two commonly consumed finfish species in both the chilled and frozen states.

## Materials and methods

### Reagents

Vitamin D3 (cholecalciferol), vitamin D3(6,19,19-d3) solution (1 mg/mL in ethanol, 97 atom % D), 25-hydroxyvitamin D3 (6,19,19-d3) solution (100 µg/mL in ethanol, 97 atom % D) and 4-phenyl-1,2,4-triazoline-3,5-dione (PTAD, 97%) were purchased from Sigma-Aldrich (St. Louis, MO). Reagent alcohol, potassium hydroxide, pyrogallol, ethyl ether anhydrous, hexanes, and acetonitrile (HPLC grade) were purchased from Fisher Scientific (Hampton, NH).

### Preparation of fish samples and irradiation

Filets of salmon and trout were provided as donations from fish vendors near Gulfport, Mississippi. All fish samples were delivered frozen to Gateway America (Gulfport, MS) for irradiation using a Gray*Star Genesis Cobalt-60 Gamma radiation source (Fig 1). Before irradiation, salmon and trout filets were dissected into four equal pieces (salmon 5.3±0.5 cm x 3.6±0.5 cm, trout 8.9±1.1 cm x 6.4±0.9 cm). Each portion was placed into an irradiation safe bag and irradiated at a different dose under chilled (0°C) and frozen (−17°C) conditions. The temperature was monitored for consistency throughout the irradiation process using Tru-temp industrial dial thermometers. One portion of three dissected fish were irradiated at doses 0, 0.5, 1, and 2 kGy for chilled conditions or doses 0, 2, 3, and 4 kGy for frozen conditions. After irradiation, samples were packaged in large coolers with dry ice and shipped overnight to Nova Southeastern University (Fort Lauderdale, FL). Once received they were transferred to a freezer and stored at −4°C until analysis.

### Saponification and vitamin D extraction

Samples were thawed overnight in a refrigerator (4°C) and the edible parts were homogenized using a stainless-steel food processor. Approximately 3–5 g of sample was transferred to a 100 mL round bottom flask and spiked with 50 µL vitamin D3(6,19,19-d$_3$) solution (2 ng/µL) internal standard, 30 mL of 2% ethanolic pyrogallic acid, and 20 mL of 50% potassium hydroxide. The flask was sealed and purged with nitrogen and allowed to stir overnight. Two extractions were performed using 30 mL of an 80:20 hexane:ether mixture and the combined extracts were washed once with 30 mL of distilled water. The solvent was evaporated using a rotary evaporator, reconstituted in 5 mL of hexane, transferred to a 20 mL vial, and evaporated to dryness. The final extract was reconstituted in 1 mL of 70% aqueous acetonitrile. The samples were shipped overnight on ice to Florida International University (Miami, FL) for LC-MS/MS analysis. Prior to analysis, 50 µL of the sample was combined with 25 µL 25-hydroxyvitamin D3 (6,19,19-d$_3$) solution (2 ng/µL) internal standard and 75 µL of PTAD (1 g/100 mL) derivatizing agent.

### LC-MS/MS analysis

LC-MS analysis was performed with an Agilent 1290 LC coupled to a 6460 QQQ-MS. The LC column used was a Zorbax reverse-phase C18 (50 x 3.0 mm, 1.8 µm) at a column temperature of 40°C. The mobile phases were (A) 5 mM ammonium formate in water with 0.1% formic acid and (B) acetonitrile with 0.1% formic acid, at a flow rate of 0.4 mL/min with

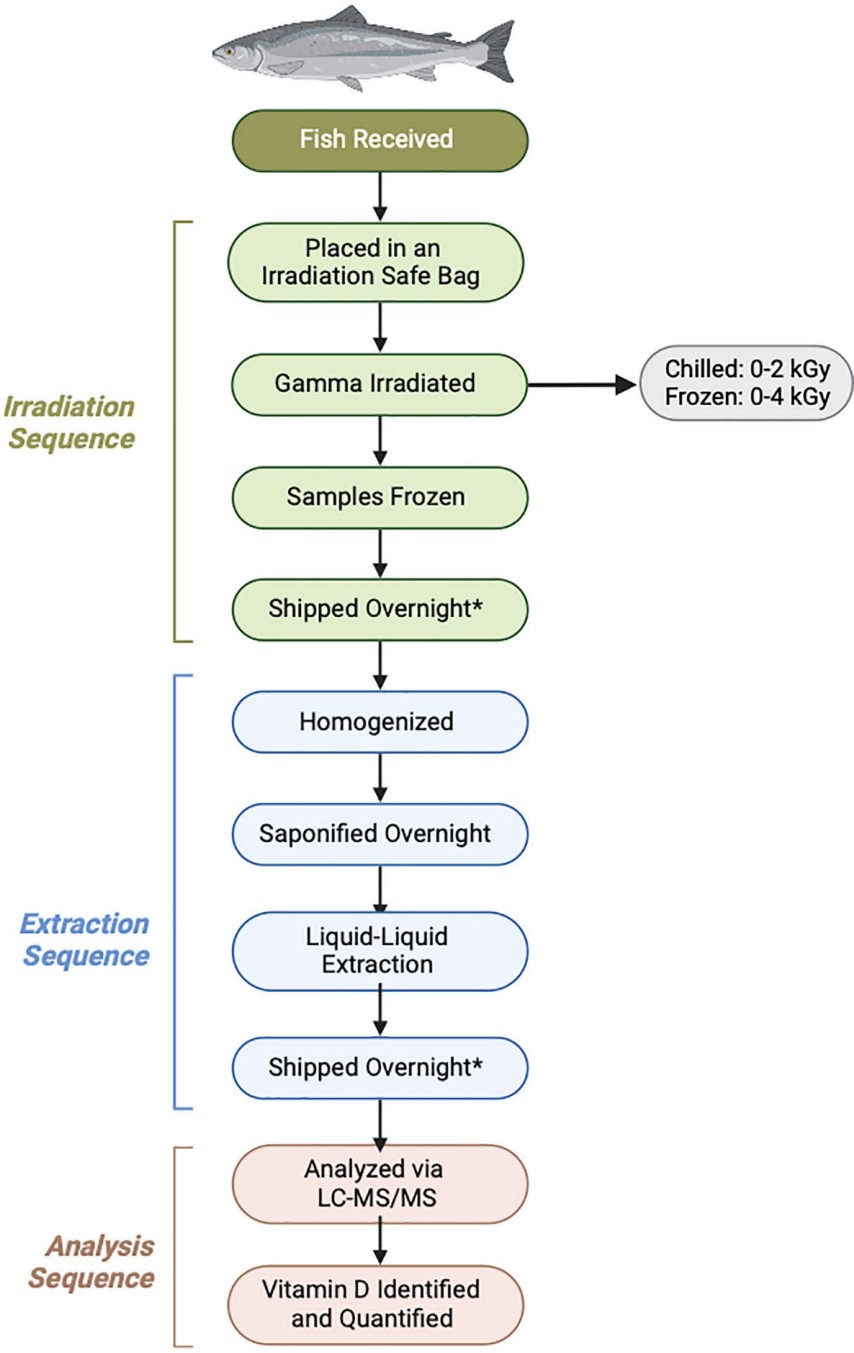

**Fig 1. Process for the gamma irradiation, extraction, and analysis of vitamin D in finfish.** Samples were stored in the freezer before, during, and after shipping.

the following gradient: hold 75% B for 3.00 min, then increase to 100% B for 2 minutes, held at 100% B for 0.5 min, and returned to 75% B for a total run time of 6 min. Sample injections were 5 µL. Detection was carried out using multiple reaction monitoring with positive ion electrospray ionization (ESI). For quantification, the transition 560.4→280.1 was

monitored for vitamin D3-PTAD and the transition 563.4→301.0 was monitored for deuterated vitamin D3-PTAD. The concentration of vitamin D3-PTAD was quantified using vitamin D3-d3-PTAD as an internal standard.

## Data analysis

Initially, changes in vitamin D content were assessed among the four portions of an individual filet. The amount of vitamin D present in an irradiated portion was compared to the amount in the unirradiated portion (control, 0 kGy) to determine the percent of vitamin D recovered in the remaining three portions following irradiation. Then the percentage of vitamin D recovered from the portions irradiated at the same dose were averaged from the three individual fish filets. Analysis was performed in JMP (version 17, Cary, NC) to determine if there was a significant change in vitamin D content following gamma irradiation when compared to the control. Normality for each finfish species and irradiation temperature was assessed using a Shapiro-Wilk test. All samples were normally distributed (salmon – chilled $P = 0.3069$; salmon – frozen, $P = 0.1907$; trout – chilled, $P = 0.727$; trout – frozen, $P = 0.8526$). Thus, statistical significance was determined using an ANOVA followed by the Tukey HSD post-hoc test for pairwise comparisons of the doses. A two-way ANOVA to assess the contribution of both finfish species and temperature on the stability of vitamin D as also performed. Here, we only used data from 2 kGy as it was the only consistent irradiation dose used for both temperatures. As this data set was found to be not normally distributaries (Shapiro Wilk; $P < 0.001$), we log transformed the data to achieve normality. We then performed a two-way ANOVA using R version 4.2.2. A significance level of 0.05 was used throughout to indicate significance.

## Results and discussion

Vitamin D was extracted by combining aspects of various previously reported protocols [28–30]. This involved alkaline saponification followed by multiple extractions. A deuterated internal standard (vitamin D3-d3) was used in all samples and calibrators. This has previously been shown to be more reliable than an external calibration curve. Additionally, vitamin D3 was derivatized with 4-phenyl-1,2,4-triazoline-3,5-dione (PTAD) to improve ionization and the signal-to-noise ratio [31]. The ratio of vitamin D3 to internal standard was used to quantify the vitamin D3 concentration in all samples. Fig 2 shows an example of a standard calibration curve and a chromatogram for the derivatized standard (vitamin D3-PTAD) and the corresponding mass spectrum for the 560.4→280.1 mass transition. The calibration curve gave high linearity ($R^2 = 0.998$), and strong responses for Vitamin D3 were observed in all samples. The limit of detection (LOD) and limit of quantitation (LOQ) of vitamin D3 for this method were 3 and 10 ng/mL, respectively.

The effect of gamma irradiation on finfish species in the chilled state was evaluated (Fig 3). Following irradiation, the amount of vitamin D in chilled salmon decreased by an average of 28%±6% for 0.5 kGy, 60%±7% for 1 kGy, and 53%±5% kGy for 2 kGy. For trout in a chilled state, the amount of vitamin D decreased by an average of 38%±10% for 0.5 kGy, 38%±29% for 1 kGy, and 62%±20% for 2 kGy. Interestingly, there was a species-specific change in the concentration of vitamin D post-irradiation. Salmon had a significant decrease in the quantity of vitamin D at 1 kGy and 2 kGy. However, when trout fillets were irradiated under the same conditions, there was no statistically significant reduction in vitamin D across all doses measured. This suggests that the chemical composition of the matrix in which vitamin D is found has different protective properties for each species; protection is significantly better in trout as compared to salmon.

To further examine this trend, the effect of irradiation on fillets of salmon and trout irradiated in the frozen condition was assessed. Owing to the previously reported protective effect of temperature on the stability of vitamins during irradiation [32], filets were irradiated at higher doses (2 kGy, 3 kGy, and 4kGy) than in the chilled state. Similar to the effect of irradiation in the chilled state, the amount of vitamin D in salmon, but not trout, declined with increasing irradiation dose (Fig 4). Following irradiation, the amount of vitamin D3 in frozen salmon decreased by an average of 36%±16% for 2 kGy, 65%±26% for 3 kGy, and 94%±2% kGy for 4 kGy. For frozen trout, the amount of vitamin D3 decreased by an average of 1%±19% for 2 kGy and increased for 3 kGy, and 4 kGy. There were significant decreases in the concentration of vitamin

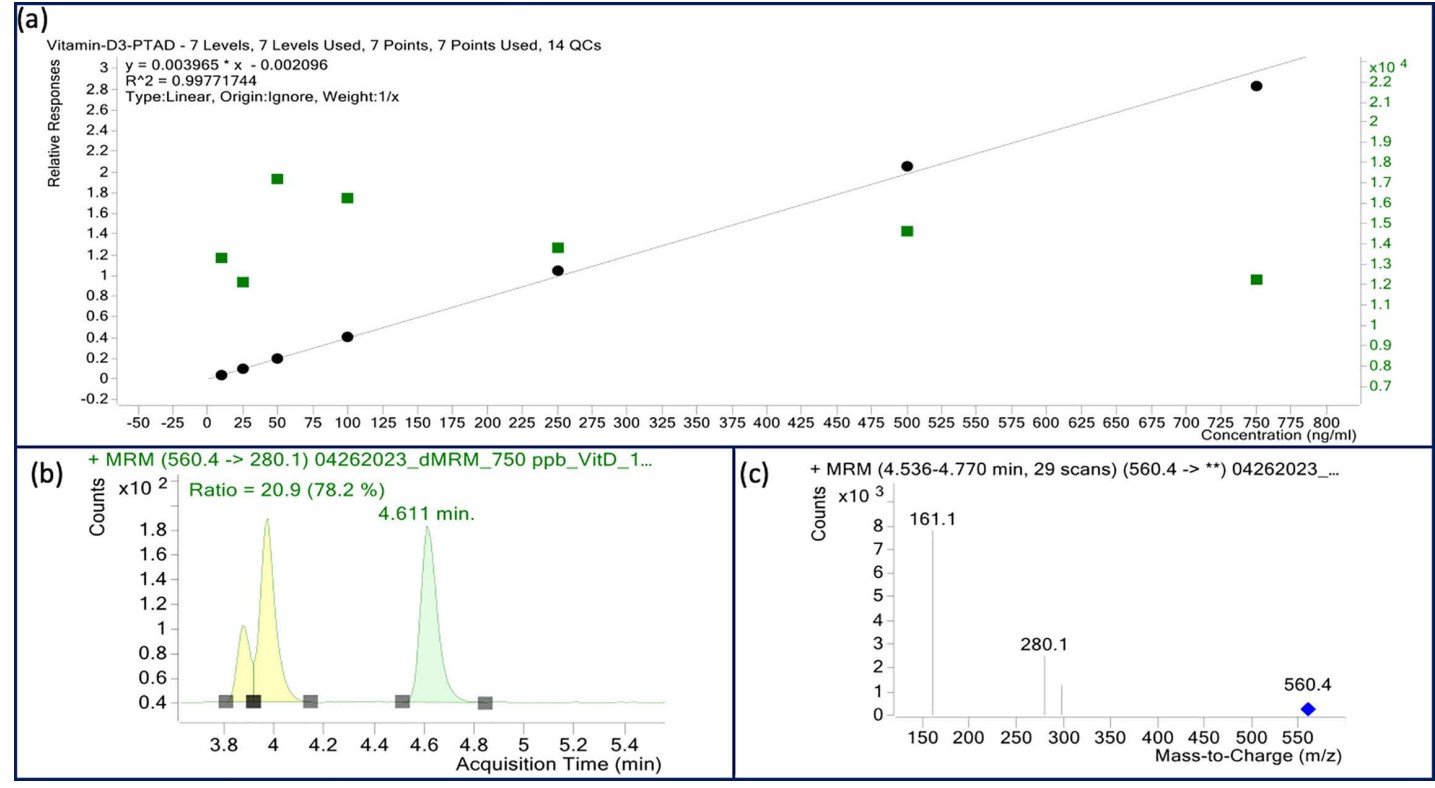

**Fig 2. LC/MS/MS results for vitamin D analysis.** (a) Standard calibration curve for the analysis of vitamin D3 ranging from 10-750 ppb spiked with 100 ppm vitamin D3-d3 internal standard. (b) Representative chromatogram for 750ppb standard and (c) mass spectrum for relevant retention time.

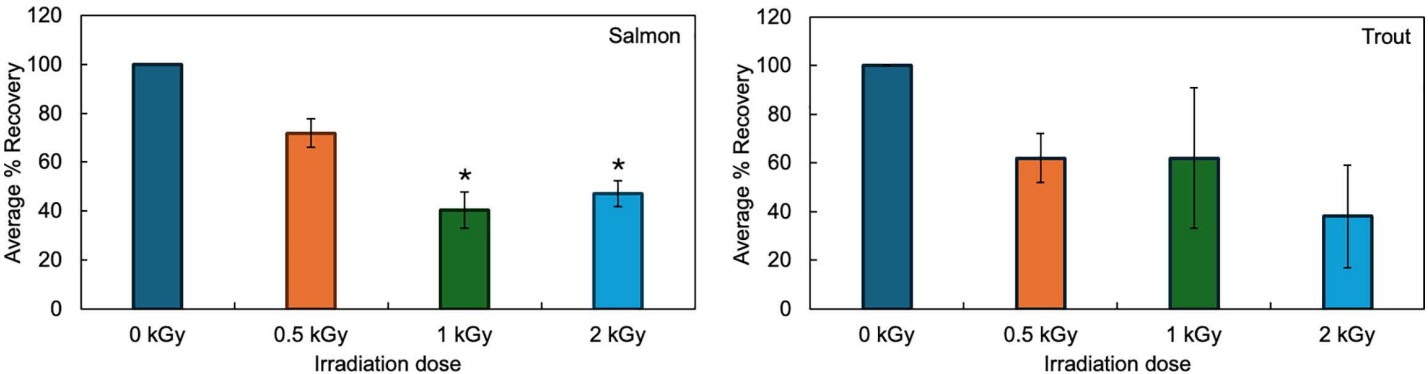

**Fig 3. The effect of irradiation on the percent recovery of vitamin D in salmon (left) and trout (right) fillets in the chilled state.** Percent recovery standardized to the unirradiated control (0 kGy). * indicates significantly different from the unirradiated control. Statistics: Salmon: ANOVA: P = 0.0013, Tukey HSD (1 kGy: P = 0.0015, 2 kGy: P = 0.0036). Trout: ANOVA: P = 0.095. Average from three different fillets, error bars = standard deviation from the mean (SEM).

D when salmon were irradiated at 3 kGy and 4 kGy. However, when trout were irradiated, there were no significant differences in the concentration of vitamin D amongst all irradiation doses examined.

We also found a statistically significant interaction between species and temperature (two-way ANOVA, P = 0.006), indicating that the effect of temperature on vitamin D stability during irradiation varies depending on the species. When

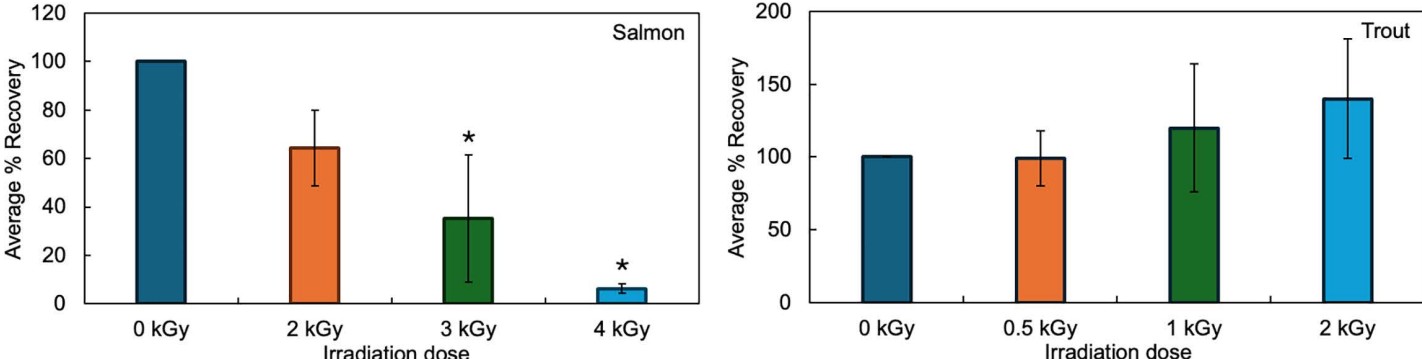

**Fig 4. The effect of irradiation on the percent recovery of vitamin D in salmon (left) and trout (right) fillets in the frozen state.** Percent recovery standardized to the unirradiated control (0 kGy). * indicates significantly different from the unirradiated control. Statistics: Salmon: ANOVA: P = 0.0045, Tukey HSD (3 kGy: P = 0.017, 4 kGy: P = 0.0039). Trout: ANOVA: P = 0.59. Average from three different fillets, with error bars = SEM.

comparing the concentration of vitamin D recovered at 2 kGy in the fresh and frozen states, but within each species, our data supports the previously noted protective effect of lower temperatures during irradiation [32]. When salmon was irradiated at 2kGy in the chilled state, a significant decrease of 53% of vitamin D was observed. However, there was no significant decrease in vitamin D when irradiated at the same dose in the frozen state; here, 64% of vitamin D remained.

Owing to the importance of vitamin D in human health, we sought to understand how the irradiation of finfish would affect its stability. We found species, temperature, and dose-dependent effects on the stability of vitamin D. Previous work found that sharpfin barracuda irradiated on ice at 5 kGy did not show a significant change in the amount of vitamin D as compared to an unirradiated control [26]. This suggests that a reduction in vitamin D is not observed at high irradiation amounts. However, reduced vitamin D due to irradiation has also been previously reported. Reductions of 5%, 10%, and 15% were observed when vitamin D3 was resuspended in either butter or cod oil and irradiated at 2.5, 5, and 10 kGy, respectively [27]. Knapp and Tappel [10] also found that irradiation at 2 kGy and 21°C reduced the concentration of vitamin D by 30% when it was resuspended in isooctane. This previous finding was consistent with our observation that when salmon was irradiated at 2 kGy, more vitamin D remained when irradiation occurred in frozen fillets as opposed to chilled fillets. Interestingly, when vitamin D was resuspended in salmon oil and irradiated at 2 kGy and 21°C [10], there was no apparent loss when irradiated under the same conditions, likely due to the protective effects of the matrix [32]. This finding is similar to our findings that showed reductions in vitamin D in salmon but not trout, which may be owing to differences in the chemical composition of the matrix and any cross-protective qualities. According to the USDA FoodData Central [33], farmed salmon and trout have similar protein content (20.4 g and 19.9 g per 100 g, respectively), but differ notably in fat and water content. Salmon contains more than twice the fat (13.4 g vs. 6.18 g) and less water (64.6 g vs. 73.8 g) than trout. We propose that these compositional differences—along with the presence of skin on one side of the trout fillets versus fully skinned salmon fillets—may help contribute to the observed species-dependent differences observed in vitamin D. How the individual components of the matrix provided cross-protective effects is largely unknown and should be studied in the future. This information could be used to determine which finfish species are likely to show a reduction in vitamin D upon irradiation.

Irradiation is often considered a method to improve food safety without affecting organoleptic properties. However, this must be evaluated for each food product, and the irradiation dose and temperature during irradiation must be considered. Warming during irradiation has been associated with a reduction in organoleptic properties [32]. Increasing irradiation dose is also associated with a reduction in organoleptic properties [34]. Accordingly, lower doses of radiation are often preferred (< 5 kGy), which have been shown to continue to inactivate between 90 and 95% of pathogens [35], thus

maintaining food safety. While the organoleptic properties of irradiated finfish were not specifically evaluated in this study, we did observe undesirable changes in color and odor. For example, when our chilled salmon samples were irradiated at doses of 2 kGy, the pink color of the salmon meat turned dull, and the fish emitted a smell akin to cooked fish. Thus, like most previous work, irradiation dose and temperature should be carefully selected to maintain the organoleptic properties of finfish while ensuring a reduction in the density of pathogenic bacteria.

Ultimately, the goal of irradiation is to reduce the quantity of pathogenic bacteria that may contaminate foods to make them safer for consumption. Importantly, the irradiation doses examined in this study, including those that did not significantly reduce vitamin D, are within and above the doses required to lead to significant reductions in pathogens most often associated with finfish, including *Vibrio* sp. For example, the $D_{10}$ value (the dose of irradiation required to achieve a 90% reduction in bacterial density) of *V. parahaemolyticus* when irradiated in saline and on ice ranges from 0.03–0.04 kGy [36]. Similar findings are reported in additional studies for *V. parahaemolyticus* [37,38,39] and additional *Vibrio* species [40,41]. *Salmonella* sp. and *Listeria* sp., both of which can be introduced during the food handling and processing phases [42–44], are also sensitive to irradiation at these doses [45,46]. For example, the density of *L. monocytogenes* in Nagli fish (*Sillago sihama*) was reduced to undetectable levels after irradiation at 3 kGy and subsequent storage for 5 days at ~1°C. However, *L. monocytogenes* was detected in the unirradiated control samples [37]. Taken together, irradiation of finfish that are potentially contaminated with the most frequently observed pathogens could potentially reduce the density of such bacteria, improving safety significantly. However, this would have to be evaluated for each finfish species studied herein in a future study.

In conclusion, we found that irradiating select finfish species could reduce the vitamin D concentration in a species, dose, and temperature-dependent manner. Together, these findings suggest that irradiation at appropriate intensities can reduce the pathogenic burden on finfish while maintaining the concentration of vitamin D.

## Acknowledgments

We thank McHenry Mauger for his assistance with the statistical analysis of the data.

## Author contributions

**Conceptualization:** Jessica S. Brown, Patricia R. Calvo, Robert P. Smith.

**Formal analysis:** Jessica S. Brown, Patricia R. Calvo, Robert P. Smith.

**Funding acquisition:** Jessica S. Brown, Patricia R. Calvo, Robert P. Smith.

**Investigation:** Jessica S. Brown, Patricia R. Calvo, Pujita Julakanti, Fatima Mohiuddin, Abdullah Basir Khan, Kemly Julien, Varun Natarajan, Leonardo B. Maya, Kaylyn A. Keith, Ryan Hollingsworth, Frank Benso.

**Methodology:** Jessica S. Brown, Patricia R. Calvo, Leonardo B. Maya, Kaylyn A. Keith, Anthony P. DeCaprio.

**Project administration:** Robert P. Smith.

**Supervision:** Jessica S. Brown, Patricia R. Calvo, Anthony P. DeCaprio, Robert P. Smith.

**Writing – original draft:** Jessica S. Brown, Patricia R. Calvo, Robert P. Smith.

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
