## [Decision Letter · Decision Letter 0]

23 Feb 2025

PONE-D-24-54557

The effect of gamma irradiation on the stability of vitamin D in select finfish species

PLOS ONE

Dear Dr. Smith,

Thank you for submitting your manuscript to PLOS ONE. After careful consideration, we have decided that your manuscript does not meet our criteria for publication and must therefore be rejected.

**Manuscript was reviewed by a member of editorial board and an external reviewer. Both find it unsuitable for publication in its current form. Findings need revisit in their experimentation, data collection and analysis and that might take much longer time, and also may not reach to a quality of manuscript to convince the conclusion drawn from the work. Authors may submit it some other journals if they wish so.**

I am sorry that we cannot be more positive on this occasion, but hope that you appreciate the reasons for this decision.

Kind regards,

Hari S. Misra, Ph.D.

Academic Editor

PLOS ONE

Reviewers' comments:

Reviewer's Responses to Questions

**Comments to the Author**

1. Is the manuscript technically sound, and do the data support the conclusions?

Reviewer #1: Yes

2. Has the statistical analysis been performed appropriately and rigorously? 

Reviewer #1: Yes

3. Have the authors made all data underlying the findings in their manuscript fully available?

Reviewer #1: Yes

4. Is the manuscript presented in an intelligible fashion and written in standard English?

Reviewer #1: Yes

5. Review Comments to the Author

**Reviewer #1: **1. The focus of this manuscript is on the study of the effect of irradiation on vitamin D in fish, but there is literature reporting that irradiation can affect the structure of the substance. Please add characterization of the structure of vitamin D and assess whether there is an effect of different irradiation doses on the structure of vitamin D and thus the function of the vitamin D.

2. Since irradiation has a certain penetration distance for sterilization, please add a flow chart of the process of actually irradiating a fish product. Because the size of the irradiation dose varies for different thicknesses of irradiated material, the structure of vitamin D may be destroyed in order to obtain a sterilizing dose.

3. Line32-42 The focus of this manuscript should be on the importance of vitamin D and the significance of using gamma radiation. If the effects of Vibrio on fish are to be illustrated, the number of Vibrio should be measured after the method is used.

4. Line43-50 List the current methods of preservation used in the transportation of fish and their advantages and disadvantages, followed by a description of the advantages of the methods used in this manuscript.

5. Line49-50 Use of the method for other articles is accompanied by corresponding literature.

6. Line126-137 Whether the effect of different gamma rays on pure vitamin D should be set.

7. Line200-201 Provide relevant literature that does not alter sensory properties.

8. Line208-209 Appropriate sensory tests including color, odor, etc. should be provided.

9. Line213-227 Will the treatment cause adverse effects and within what range will it not cause adverse effects with references.

6. PLOS authors have the option to publish the peer review history of their article (what does this mean?). If published, this will include your full peer review and any attached files.

Reviewer #1: No

- - - - -

---

## [Author Response · Author response to Decision Letter 1]

20 Mar 2025

Dear Dr. Mitra,

While we would like to thank you and the reviewer for evaluating our manuscript, we believe that you and the reviewer have made fundamental errors in your assessment, not abided by the review standards set forth by PLoS One, and have simply asked us ‘to do more work’ that is clearly outside the scope of our manuscript.

PLoS One evaluates based on technical rigor and not novelty. As listed on their website, the criteria for review are the following:

1. The study presents the results of original research. Our work and data are entirely novel as they detail the effects of irradiation on vitamin D in commercially important finfish species.

2. Results reported have not been published elsewhere. Our results have not been published elsewhere. They are entirely novel.

3. Experiments, statistics, and other analyses are performed to a high technical standard and are described in sufficient detail. Not a single comment below indicates that the technical aspects of the proposal are flawed. We have applied appropriate statistics, including correcting for multiple comparisons, have sufficiently high replication, our experiments and analyses were performed according to industry standards.

4. Conclusions are presented in an appropriate fashion and are supported by the data. Our conclusion, as indicated in the discussion and abstract, is that irradiation can affect the stability of vitamin D in a dose-dependent, temperature-dependent, and fish species-specific data. This conclusion is well supported by the data presented in Figures 2 and 3, and is supported by a statistical analysis. Further, not a single comment below challenged the fundamental conclusions made in our manuscript.

5. The article is presented in an intelligible fashion and is written in standard English. Our manuscript was written and edited by authors with over 50 publications. Not a single comment below indicates that our manuscript needs to be better written; it only indicates that more work needs to be done for, quite frankly, the sake of asking for more work as it is clearly outside of the scope of the manuscript.

6. The research meets all applicable standards for the ethics of experimentation and research integrity. Not a single comment below indicated any issues with ethics. In fact, we supplied all of our raw data for the reviewers to analyze, including calibration curves (Fig. 1). This far exceeds the standard of the community.

7. The article adheres to appropriate reporting guidelines and community standards for data availability. Not a single comment below indicates that our data is not reported correctly or does not meet the community standard of availability. All of our raw data is indeed included in the manuscript.

As further evidence that both you and reviewer did not follow the standard of review and assessment as set forth by PLoS One, the reviewer indicated that all of the above was fine in their evaluation. Directly from the reviewer report:

Comments to the Author

1. Is the manuscript technically sound, and do the data support the conclusions?

Reviewer #1: Yes

2. Has the statistical analysis been performed appropriately and rigorously?

Reviewer #1: Yes

3. Have the authors made all data underlying the findings in their manuscript fully available?

Reviewer #1: Yes

4. Is the manuscript presented in an intelligible fashion and written in standard English?

Reviewer #1: Yes

Thus, all of the criteria required for publication in PLoS One have been met. What remains are comments from the reviewer that lack a scientific basis and clarity, are outside the scope of our manuscript, or are so minor in their scope that 10 minutes in a document editing system could fix the issue. Certainly, none of these comments, either alone or as a synthesis, warrants a rejection as we have clearly met the review criteria for PLoS One as indicated by the reviewer themselves in the report above.

Our responses to the reviewer comments are below.

1. The focus of this manuscript is on the study of the effect of irradiation on vitamin D in fish, but there is literature reporting that irradiation can affect the structure of the substance. Please add characterization of the structure of vitamin D and assess whether there is an effect of different irradiation doses on the structure of vitamin D and thus the function of the vitamin D.

This is outside of the scope of our manuscript. The central focus of our manuscript is the stability of vitamin in finfish after irradiation. As clearly indicated in the introduction:

“While irradiation represents an established strategy that could reduce the incidence of

pathogens found in finfish, it remains unclear as to how irradiation impacts the stability of vitamin D in finfish tissue of commercial species.”

This is exactly what we have studied and precisely what we have reported on.

Our central conclusion, as clearly indicated in the abstract is:

“Overall, our results indicate that irradiation of finfish may not reduce vitamin D concentration when applied at dosages of 0.5 kGy and 2 kGy or less in chilled and frozen conditions, respectively.”

And, from our discussion:

“In conclusion, we found that irradiating select finfish species could reduce the vitamin D concentration in a species, dose, and temperature-dependent manner. Together, these findings suggest that irradiation at appropriate intensities can reduce the pathogenic burden on finfish while maintaining the concentration of vitamin D.”

We make no claim on the structure of vitamin D, nor does it matter to any single conclusion made in our manuscript. This comment does not impact the technical rigor or conclusions made in the manuscript and is thus entirely unwarranted and outside the scope of our manuscript.

The reviewer is correct that the effects of ionizing radiation on vitamin D have been previously studied. In fact, it is commonly known that to produce vitamin D, ionizing radiation is required. The steps in this process have been well documented 1. Accordingly, there is no reason to reproduce this data in this manuscript.

1

Finally, Vitamin D was accurately identified by comparing its retention time to a pure chemical standard and confirming the match through molecular and fragment ion patterns in the mass spectral data. This community standard for measuring vitamin D has been used multiple times before 2-12. We have applied the most rigorous methods to detect vitamin D.

Taken together, this comment is a) outside the scope of the manuscript, b) does not challenge the fundamental analysis or conclusions of the manuscript, and c) provides no value to the manuscript if performed.

2. Since irradiation has a certain penetration distance for sterilization, please add a flow chart of the process of actually irradiating a fish product. Because the size of the irradiation dose varies for different thicknesses of irradiated material, the structure of vitamin D may be destroyed in order to obtain a sterilizing dose.

We have included both the dosimetry data and a figure showing the irradiation process (Figure 1). This new material can be found in the methods section.

However, the second sentence of the reviewer’s comment is entirely unclear. We have no interest and make no claim on sterilizing doses. That is not a conclusion of our manuscript, nor is it a goal. Determining a sterilizing will have not change the fundamental conclusions of our manuscript, nor will it add anything to the manuscript within its scope.

Further to that, many, if not all, papers that have looked at the stability of vitamins and nutrients in finfish or other matrices do not identify sterilizing dose 13-30. Why? Because it is entirely unnecessary and outside of the scope of this and the work of others.

We also note that sterilizing doses for multiple finfish species have been established elsewhere 31. Thus, this work is superfluous to what has already been produced by the community.

In summary, we added a flow chart and dosimetry data. However, determining a sterilizing dose is completely unnecessary within the scope of the manuscript and will have no fundamental impact on our conclusion or data analysis.

3. Line32-42 The focus of this manuscript should be on the importance of vitamin D and the significance of using gamma radiation. If the effects of Vibrio on fish are to be illustrated, the number of Vibrio should be measured after the method is used.

We included this information in our original manuscript to highlight the use of gamma irradiation on fish due to concerns about pathogenic bacteria, Vibrio being chief amongst them. Since numerous studies have already demonstrated that gamma irradiation effectively eliminates Vibrio in fish, we did not repeat these experiments. In light of the reviewer’s comment, we have removed this paragraph.

4. Line 43-50 List the current methods of preservation used in the transportation of fish and their advantages and disadvantages, followed by a description of the advantages of the methods used in this manuscript.

Our primary focus is on vitamin D stability and irradiation of finfish, not on all preservation and transportation methods of finfish; that material belongs in a review article. Adding this information will only muddy the waters in terms of the central focus of our manuscript, will not fundamentally enhance readability, or will not change our analysis or conclusions. In addition, many manuscripts have been published on the irradiation of foods, and the vast majority do not summarize this information as it is superfluous and obscures the central message of these publications 2-12.

5. Line 49-50 Use of the method for other articles is accompanied by corresponding literature.

It is not clear what the reviewer is asking for here. We think that they might be asking for references indicating approval for the irradiation of foods in the United States. We have included these references.

6. Line 126-137 Whether the effect of different gamma rays on pure vitamin D should be set.

This has already been determined previously, as noted in our original manuscript.

“Previous work performed on whole extracts of finfish32, vitamin D resuspended in organic solvents32, and chicken25 has yielded somewhat conflicting results. On the one hand, some studies show that irradiation does not impact the concentration of vitamin D 33. On the other, some previous work has shown that irradiation can reduce the concentration of vitamin D 5,32”

For example, previous work has assessed the stability of pure vitamin D in benzene, ethanol, cod oil, liver oil, and butter 5,32. This synthesis of work has clearly established the effects of irradiation on pure vitamin D. Our work establishes the impact of irradiation on vitamin D in finfish and is entirely different from this previous work.

Therefore, why repeat this work? It would not fundamentally alter our manuscript's technical rigor, conclusions, or anything else. Once again, this is a reviewer comment that asks for more work for the sake of asking for more work. It is entirely irrelevant to our manuscript.

Moreover, nearly all other manuscripts investigating the stability of a vitamin or nutrient in foods do not look at the purified product 13-27. Why? Because, as clearly stated in our introduction, the matrix of the food can determine the stability 34.

“However, to date, not a single study has investigated the stability of vitamin D in fresh, commercially important finfish species. This is important to consider as the composition of the chemical matrix where vitamin D is found may react differently to irradiation and may provide protection against any loss.”

The premise of the above statement has been established in multiple studies and in multiple foods (reviewed in 34), but not commercially important finfish, which we are examining here as demonstrated by our specific questions outlined in the introduction

“While irradiation represents an established strategy that could reduce the incidence of pathogens found in finfish, it remains unclear as to how irradiation impacts the stability of vitamin D in finfish tissue of commercial species.”

Thus, not only is the proposed work entirely outside of the scope of the manuscript, but it is entirely contradictory to the specific question we asked, as indicated in the introduction.

To clarify that this work has already been completed, the introduction now reads:

“Previous work performed on whole extracts of finfish32, purified vitamin D resuspended in organic solvents32, and chicken25 has yielded somewhat conflicting results. On the one hand, some studies show that irradiation does not impact the concentration of vitamin D 33. On the other, some previous work has shown that irradiation can reduce the concentration of vitamin D 5,32”

7. Line 200-201 Provide relevant literature that does not alter sensory properties.

It is not clear what the reviewer is requesting here. However, we think that they are asking to provide a reference about the effects of irradiation on organoleptic properties. While this was discussed in depth and with references in the paragraph, we have removed the paragraph in response to the previous comment.

8. Line 208-209 Appropriate sensory tests including color, odor, etc. should be provided.

This is entirely outside the scope of our manuscript. This suggested work will have absolutely no impact on our technical rigor or fundamental conclusions. Moreover, multiple other studies that have examined the effect of irradiation on nutrients and vitamins do not do this 14,17,18,25-30.

The reviewer is simply asking for more work for the sake of asking for more work. We make absolutely no claim on the organoleptic properties of finfish; simply that we noticed changes and that they need to be evaluated elsewhere and carefully chosen. As indicated in our discussion:

“Irradiation is often considered a method to improve food safety without affecting organoleptic properties. However, this must be evaluated for each food product, and the irradiation dose and temperature during irradiation must be considered. Warming during irradiation has been associated with a reduction in organoleptic properties34. Increasing irradiation dose is also associated with a reduction in organoleptic properties35. Accordingly, lower doses of radiation are often preferred (< 5 kGy), which have been shown to continue to inactivate between 90 and 95% of pathogens36, thus maintaining food safety. While the organoleptic properties of irradiated finfish were not specifically evaluated in this study, we did observe undesirable changes in color and odor. For example, when our chilled salmon samples were irradiated at doses of 2 kGy, the pink color of the salmon meat turned dull, and the fish emitted a smell akin to cooked fish. Thus, like most previous work, irradiation dose and temperature should be carefully selected to maintain the organoleptic properties of finfish while ensuring a reduction in the density of pathogenic bacteria.”

In response to the reviewers' comment, we have removed this entire paragraph.

9. Line 213-227 Will the treatment cause adverse effects and within what range will it not cause adverse effects with references.

It is unclear what the reviewer is requesting. We think that they might be asking about organoleptic properties. Accordingly, we have added information about previous studies that have irradiated finfish within the doses we used and examined organoleptic properties post-irradiation. This can be found in the second last paragraph of the discussion.

In conclusion, the reviewer made multiple fundamental errors in assessing our manuscript. The majority of the comments are outside our paper's scope, do not fundamentally affect the analysis, conclusions, or clarity of our manuscript, are nearly impossible to interpret, and are essentially comments asking for more work for the sake of asking for more work. Simply doing more experiments that do not impact the fundamental technical rigor, significance, or conclusions of a manuscript owing to the whims of a reviewer is not a review criterion established by PLoS One.

Literature Cited

1 Bikle, D. D.

---

## [Decision Letter · Decision Letter 1]

20 Jun 2025

PONE-D-24-54557R1The effect of gamma irradiation on the stability of vitamin D in select finfish speciesPLOS ONE

Dear Dr. Smith

Thank you for submitting your manuscript to PLOS ONE. As editor, After careful consideration, we feel that it has merit but does not fully meet PLOS ONE’s publication criteria as it currently stands. Therefore, we invite you to submit a revised version of the manuscript that addresses the points raised during the review process.Please submit your revised manuscript by June 30. If you will need more time than this to complete your revisions, please reply to this message or contact the journal office at plosone@plos.org. Please include the following items when submitting your revised manuscript:

We look forward to receiving your revised manuscript.

Kind regards,

Shafaq Fatima

Academic Editor

PLOS ONE

Journal Requirements:

“Robert P Smith received funding from the Seafood Industry Research Fund. However, the funding does not cover the cost of publication. “

“This work was supported by the Seafood Industry Research Fund through the National Fisheries Institute.”

“Robert P Smith received funding from the Seafood Industry Research Fund. However, the funding does not cover the cost of publication.”

“Robert Smith is an authors on a petition to the US FDA that seeks the approval of irradiation for finfish and flatfish.”

Additional Editor Comments (if provided):

Reviewers' comments:

Reviewer's Responses to Questions

**Comments to the Author**

1. If the authors have adequately addressed your comments raised in a previous round of review and you feel that this manuscript is now acceptable for publication, you may indicate that here to bypass the “Comments to the Author” section, enter your conflict of interest statement in the “Confidential to Editor” section, and submit your "Accept" recommendation.

Reviewer #2: (No Response)

2. Is the manuscript technically sound, and do the data support the conclusions?

Reviewer #2: Yes

3. Has the statistical analysis been performed appropriately and rigorously? 

Reviewer #2: No

4. Have the authors made all data underlying the findings in their manuscript fully available?

Reviewer #2: Yes

5. Is the manuscript presented in an intelligible fashion and written in standard English?

Reviewer #2: No

6. Review Comments to the Author

Reviewer #2: Abstract

Q.1

What was the duration of the study or trial period during which samples were monitored or stored?

Q.2

How many samples were used per species and per treatment condition (irradiation dose, temperature)?

Introduction

Q.3

Furthermore, irradiating chicken, beef, lettuce, and crustaceans is already commonplace in the United States. Add references (Line 49-50).

Q.4

In 2021, Americans consumed an average of 20.5 pounds of finfish and shellfish, representing the highest average consumption weight reported to date. How much of this consumption is specifically attributable to finfish alone? Is it appropriate to generalize this statistic to support an argument focused only on finfish? (Line 27)

Q.5

Irradiation causes DNA damage to kill bacteria, but how does this vary across different bacterial genera mentioned (e.g., Vibrio vs. Clostridium)? Are there species-specific resistance thresholds to irradiation? (Line 45-46)

Q.6

What is the rationale for testing both chilled and frozen conditions?

Methodology

Q.7

How many individual fish were used per species for the entire experiment? The method only mentions "three dissected fish. (Line 75)

Q.8

Were all pieces taken from the same fish used as replicates for each dose, or were they from different fish? If from the same fish, how do the authors justify the use of pseudo-replication in the statistical analysis?

Q.9

Why were different dose ranges used for chilled (0–2 kGy) and frozen (0–4 kGy) conditions? Is there literature or preliminary data justifying this discrepancy? Add reference (Line 83)

Q.10

How were the chilled (0°C) and frozen (-17°C) conditions monitored and maintained during irradiation? (Line 80)

Q.11

What was the rationale for using vitamin D3-d3 as the internal standard at 50 µL (2 ng/µL)? Was the spike level optimized to fall within the linear range of the LC-MS/MS?

Q.12

Samples were stored at –4°C after irradiation. This is an unusually high "freezer" temperature for sample preservation. Could vitamin D degradation have occurred prior to extraction and analysis? (Line 88)

Q.13

Was the assumption of homoscedasticity (equal variance) checked before applying ANOVA? If heterogeneity of variance was present, why was ANOVA used rather than a non-parametric test or transformation?

Results and Discussion

Q.14

The data are reported as mean ± SEM. How many technical replicates per biological replicate were run to ensure analytical precision?

Q.15

Why does 1 kGy in chilled salmon show a greater loss (60%) than 2 kGy (53%)? (Line 144)

Q.16

Author hypothesized a matrix effect protects vitamin D in trout but not salmon. But no lipidomic, proteomic, or moisture content comparisons made between the two species to support this claim.

Q.17

2 kGy dose in frozen salmon retained more vitamin D than the chilled equivalent, what are the implications of partial thawing or ice crystal formation during irradiation?

Q.18

Authors should perform Two-way ANOVA to compare the effects of species and temperature simultaneously? A full factorial analysis could better reveal interaction effects.

Q.19

Author mentioned that previous work showing no vitamin D loss at 5 kGy in sharpfin barracuda. Could this species-specific protection due to matrix effects, or are methodological differences?

Q.20

Author must assess the microbial load post-irradiation to confirm pathogen reduction. If not, how this manuscript confidently mentioned that irradiation at appropriate intensities can reduce the pathogenic burden on finfish while maintaining the concentration of vitamin D? (Line 230).

7. PLOS authors have the option to publish the peer review history of their article (what does this mean?). If published, this will include your full peer review and any attached files.

Reviewer #2: No

---

## [Author Response · Author response to Decision Letter 2]

2 Jul 2025

We sincerely thank the reviewers for their thoughtful and constructive feedback on our manuscript. We have carefully considered each comment and have revised the manuscript accordingly. Below, we have copied each reviewer comment in full and provided our detailed response directly beneath each one. Changes made to the manuscript are indicated where applicable, and line numbers have been provided to identify the location of each change. Importantly, the line numbers reflect lines in the marked up version of the manuscript, and not the clean version of the manuscript.

Reviewer #2:

Abstract

Q.1 What was the duration of the study or trial period during which samples were monitored or stored?

The research was performed over a nine-month period.

• Irradiation: Late October 2022

• Vitamin D extraction

o Salmon, chilled: Late November to late December 2022

o Salmon, frozen: Late October to late November 2022

o Trout, frozen: January 2023

o Trout, chilled: February to mid-March 2023

• LC-MS analysis

o Salmon, chilled: Late January 2023

o Salmon, frozen: Late January 2023

o Trout, frozen: Late January 2023

o Trout, chilled: Late June 2023

However, we feel as though this timeline is not relevant to the findings or conclusions of our manuscript. Thus, we have not modified the abstract.

Q.2 How many samples were used per species and per treatment condition (irradiation dose, temperature)?

Three filets (or samples) were used for each species and temperature combination (e.g., three filets of chilled salmon), totaling 12 filets for the four combinations. Each filet was dissected into 4 portions, and each portion was irradiated with a different dose of gamma irradiation. We added a clarifying sentence to the abstract (Lines 132-134).

Introduction

Q.3 Furthermore, irradiating chicken, beef, lettuce, and crustaceans is already commonplace in the United States. Add references (Line 49-50).

A reference from the FDAs approval of gamma irradiation was added to the manuscript (Line 179).

Q.4 In 2021, Americans consumed an average of 20.5 pounds of finfish and shellfish, representing the highest average consumption weight reported to date. How much of this consumption is specifically attributable to finfish alone? Is it appropriate to generalize this statistic to support an argument focused only on finfish? (Line 27)

We acknowledge that the reported figure of 20.5 pounds includes both finfish and shellfish; however, we chose to include this statistic because it is a widely reported and an authoritative measure of U.S. seafood consumption trends. While it does not distinguish between finfish and shellfish, it effectively illustrates the broader point that seafood consumption is increasing in the United States.

Q.5 Irradiation causes DNA damage to kill bacteria, but how does this vary across different bacterial genera mentioned (e.g., Vibrio vs. Clostridium)? Are there species-specific resistance thresholds to irradiation? (Line 45-46)

The references included in the manuscript were intended to demonstrate that the use of gamma irradiation to eliminate bacterial pathogens is a well-established and accepted practice. While we recognize that bacterial resistance to irradiation can vary by species and genus, a detailed examination of species-specific resistance thresholds was beyond the scope of our study.

Nevertheless, our primary objective was to evaluate whether gamma irradiation adversely affects the nutritional components of fish, rather than to assess its differential efficacy across bacterial genera. A sentence was added to the introduction to specify this objective (Lines 180-183)

Q.6 What is the rationale for testing both chilled and frozen conditions?

Commercial fish products are sold either fresh (chilled) or frozen and we sought to test the stability of vitamin D for both types of products. A sentence was added to the introduction to explain our choice for irradiating at chilled and frozen conditions (Lines 190-195).

Additionally, it has also been well-established that reducing the temperature of irradiation can reduce the effects of irradiation on changes in nutrition and microbial load. Thus, we chose to examine the impacts of irradiation at these two temperatures. Two sentences were added to the introduction that mentions this occurrence and why we chose two ranges of doses for chilled and frozen samples (Lines 173-175).

Methodology

Q.7 How many individual fish were used per species for the entire experiment? The method only mentions "three dissected fish. (Line 75)

Three filets (or samples) were used for each species and temperature combination (e.g., three filets of chilled salmon), totaling 12 filets for the four combinations. Each filet was dissected into 4 portions, and each portion was irradiated with a different dose of gamma irradiation.

Q.8 Were all pieces taken from the same fish used as replicates for each dose, or were they from different fish? If from the same fish, how do the authors justify the use of pseudo-replication in the statistical analysis?

To account for biological variability among fish samples and to standardize comparisons across irradiation doses, we selected three filets from three different individual fish and dissected each into four portions. Each piece was then subjected to a different irradiation dose. This approach allowed us to monitor relative changes in vitamin D concentration within the same filet, using the non-irradiated portion as the control (set at 100%). By doing so, we minimized inter-sample variability and ensured that observed differences were attributable to irradiation rather than natural variation among individual fish. This has been clarified in the Data Analysis subsection of the Materials and Methods section (Lines 288-291).

Q.9 Why were different dose ranges used for chilled (0–2 kGy) and frozen (0–4 kGy) conditions? Is there literature or preliminary data justifying this discrepancy? Add reference (Line 83)

Previous studies show that gamma irradiation is less effective at killing bacteria at lower temperatures, requiring higher doses at subfreezing conditions [1]. These ranges are also consistent with previous work that has examined the impacts of irradiation on finfish, as well as irradiation doses currently used to treat finfish in international markets. Two sentences were added to the introduction that mentions this occurrence and why we chose two ranges of doses for chilled and frozen samples (Lines 173-175).

Q.10 How were the chilled (0°C) and frozen (-17°C) conditions monitored and maintained during irradiation? (Line 80)

All irradiation was performed at Gateway America, which routinely irradiates food for commercial distribution. During the irradiation process, probes that monitor temperature and dose are inserted amongst the irradiated samples. All irradiation occurs in a large, refrigerated warehouse, which assists to ensure that fluctuations in temperature do not occur. We have clarified this in the Methods section (Lines 226-227)

Q.11 What was the rationale for using vitamin D3-d3 as the internal standard at 50 µL (2 ng/µL)? Was the spike level optimized to fall within the linear range of the LC-MS/MS?

During initial method development, a larger volume of internal standard (IS) was added to each sample. However, we observed that this produced a disproportionately large IS signal relative to the vitamin D3 signal in fish extracts, which could interfere with accurate quantitation. As a result, we optimized the IS spike volume to 50 µL at 2 ng/µL to ensure that the IS concentration was comparable to the analyte concentration in typical samples. This adjustment provided a more balanced signal, minimized the risk of ion suppression or detector saturation, and ensured accurate, reproducible quantitation within the linear dynamic range of the method.

Q.12 Samples were stored at –4°C after irradiation. This is an unusually high "freezer" temperature for sample preservation. Could vitamin D degradation have occurred prior to extraction and analysis? (Line 88)

Previous studies have shown that vitamin D3 is more stable at lower temperatures, with significantly greater stability reported at 4 °C compared to 27 °C [2]. Thus, we are confident that storing our samples at –4 °C, a subfreezing temperature, did not lead to degradation of vitamin D3 prior to extraction and analysis. Although –4 °C is slightly higher than typical ultra-low freezer settings (e.g., –20 °C), it is still below the threshold at which vitamin D degradation has been reported under standard storage conditions.

Q.13 Was the assumption of homoscedasticity (equal variance) checked before applying ANOVA? If heterogeneity of variance was present, why was ANOVA used rather than a non-parametric test or transformation?

As noted in our original submission the distribution of all vitamin D for each finfish species and irradiation temperature was assessed using a Shapiro-Wilk test for normality. Thus, an ANOVA followed by a Tukey HSD is highly appropriate. We have updated our manuscript to include P values for each finfish species and temperature for the Shapiro-Wilk test in the Methods section (Lines 293-296).

Results and Discussion

Q.14 The data are reported as mean ± SEM. How many technical replicates per biological replicate were run to ensure analytical precision?

To ensure analytical precision, randomly selected samples were run in duplicate as technical replicates. This approach allowed us to confirm consistency in measurement across runs without significantly depleting the limited sample volume available for analysis. However, because technical replicates were not consistently run for each sample, we only used biological replicates for our analysis.

Q.15 Why does 1 kGy in chilled salmon show a greater loss (60%) than 2 kGy (53%)? (Line 144)

While it would appear that 1 kGy showed a great loss as compared to 2 kGy, these data points are in fact not different from each other. A two-tailed t-test comparing vitamin D isolated from chilled salmon irradiated at 1kGy and 2kGy did not show significance (P = 0.44). The averages are also well within the error range as noted by the overlapping error bars.

Q.16 Author hypothesized a matrix effect protects vitamin D in trout but not salmon. But no lipidomic, proteomic, or moisture content comparisons made between the two species to support this claim.

Performing lipidomic, proteomic or moisture content analysis is beyond the scope of this study. However, according to the USDA FoodData Central [3], farmed salmon and trout have similar protein content (20.4 g and 19.9 g per 100 g, respectively), but differ notably in fat and water content. Salmon contains more than twice the fat (13.4 g vs. 6.18 g) and less water (64.6 g vs. 73.8 g) than trout. While we did not directly measure these properties in our samples, we proposed that such compositional differences—along with the presence of skin on one side of the trout fillets versus fully skinned salmon fillets—may help explain the species-dependent differences observed in vitamin D stability following irradiation. These details have been added to the Results and Discussion section (Lines 417-422).

Q.17 2 kGy dose in frozen salmon retained more vitamin D than the chilled equivalent, what are the implications of partial thawing or ice crystal formation during irradiation?

We did not observe thawing or ice crystal formation during irradiation, as the temperature was kept constant. It is expected that Vitamin D stability is improved at lower temperatures, which has been previously demonstrated for other compounds during irradiation.

Q.18 Authors should perform Two-way ANOVA to compare the effects of species and temperature simultaneously? A full factorial analysis could better reveal interaction effects.

Unfortunately, we cannot run a two-way ANOVA across our entire data set as only 0 kGy and 2kGy were the only consistently tested irradiation doses tested between both temperatures.

We did, however, perform a two-way ANOVA using the 2 kGy data. Because a Shapiro-Wilk test performed on only the 2kGy indicated that it was not normally distributed (P < 0.0001), we log-transformed the data to achieve normality. We then performed a two way ANOVA considering both species and temperature. Our findings indicate that there is a significant effect of both finfish species and temperature on the stability of vitamin D (P = 0.006). This analysis has been added to the Materials and Methods section (Lines 297-315) and the results included in the Results and Discussion section (Lines 384-386).

Q.19 Author mentioned that previous work showing no vitamin D loss at 5 kGy in sharpfin barracuda. Could this species-specific protection due to matrix effects, or are methodological differences?

It could be either.

The methods used in the article noted above were not well described and simply stated that an 1990 version of the AOAC official method was used. Thus, it is no clear exactly clear what they performed and how it might differ from our extraction and analysis methods. Additionally, the barracuda was sun-dried and had a moisture content of 24.6%, while our samples were raw and had water content of 64.6% for salmon and 73.8% for trout [3].

We also note that it could be a species specific difference due to matrix effects. For example, non-irradiated sharpfin barracuda has 7.09% lipids, 7.09g/100g or lipids. This contrasts to 13.4 g vs. 6.18 g of fat per 100g observed in salmon and trout, respectively [3].

Q.20 Author must assess the microbial load post-irradiation to confirm pathogen reduction. If not, how this manuscript confidently mentioned that irradiation at appropriate intensities can reduce the pathogenic burden on finfish while maintaining the concentration of vitamin D? (Line 230).

It was not the goal of this work to demonstrate the microbial reduction potential of irradiation, as it has been demonstrated extensively in other studies [4-11]. This literature supports that doses much lower than the doses used in this study effectively reduce or eliminate microbial load. This study was solely focused on the effects of irradiation on Vitamin D.

We have modified to statement in the results/discussion section to make it clear that our claim is based on previous literature, and not examined in this manuscript. We have also noted that this should be examine in a future study (Lines 451-456)

References

1. Radomyski T, Murano EA, Olson DG, Murano PS. Elimination of pathogens of significance in food by low-dose irradiation: a review. Journal of food protection. 1994;57(1):73–86.

2. Zareie M, Abbasi A, Faghih S. Influence of storage conditions on the stability of vitamin D3 and kinetic study of the vitamin degradation in fortified canola oil during the storage. Journal of Food Quality. 2021;2021(1):5599140.

3. United States Department of Agriculture ARS. FoodData Central 2019 [cited 2025]. Available from: fdc.nal.usda.gov.

4. Dion P, Charbonneau R, Thibault C. Effect of ionizing dose rate on the radioresistance of some food pathogenic bacteria. Canadian journal of microbiology. 1994;40(5):369–74.

5. Ahmed IO, Alur MD, Kamat AS, Bandekar JR, Thomas P. Influence of processing on the extension of shelf‐life of Nagli‐fish (Sillago sihama) by gamma radiation. International journal of food science & technology. 1997;32(4):325–32.

6. Jakabi M, Gelli DS, Torre JC, Rodas MA, Franco BD, Destro MT, et al. Inactivation by ionizing radiation of Salmonella enteritidis, Salmonella infantis, and Vibrio parahaemolyticus in oysters (Crassostrea brasiliana). Journal of food protection. 2003;66(6):1025–9.

7. Matches J, Liston J. Radiation destruction of Vibrio parahaemolyticus. Journal of Food Science. 1971;36(2):339–40.

8. Ama AA, Hamdy M, Toledo R. Effect of heating, pH and thermoradiation on inactivation of Vibrio vulnificus. Food Microbiology. 1994;11(3):215–27.

9. Torres Z, Kahn G, Vivanco M, Guzman G, Bernuy B. Shelf-life extension and decontamination of fish fillets (Trachurus picturatus murphyi and Mugil cephalus) and shrimp tails (Penaeus vannamei) inoculated with toxigenic Vibrio cholerae O1 El Tor using gamma radiation. Irradiation to control Vibrio infection from consumption of raw seafood and fresh produce I

---

## [Decision Letter · Decision Letter 2]

3 Sep 2025

PONE-D-24-54557R2The effect of gamma irradiation on the stability of vitamin D in select finfish speciesPLOS ONE

Dear Dr. Smith

Thank you for submitting your manuscript to PLOS ONE. After careful consideration, we feel that it has merit but does not fully meet PLOS ONE’s publication criteria as it currently stands. Therefore, we invite you to submit a revised version of the manuscript that addresses the points raised during the review process. Please address the following comment from reviewer:I recommend a final check to ensure that all references cited in the text are properly included in the Literature Cited section, and that the formatting is consistent throughout.

We look forward to receiving your revised manuscript.

Kind regards,

Shafaq Fatima

Academic Editor

PLOS ONE

Journal Requirements:

Reviewer #2: The authors have carefully addressed all previous comments and concerns, and the revised manuscript now appears suitable for publication. I recommend a final check to ensure that all references cited in the text are properly included in the Literature Cited section, and that the formatting is consistent throughout.

**Comments to the Author**

1. If the authors have adequately addressed your comments raised in a previous round of review and you feel that this manuscript is now acceptable for publication, you may indicate that here to bypass the “Comments to the Author” section, enter your conflict of interest statement in the “Confidential to Editor” section, and submit your "Accept" recommendation.

Reviewer #2: All comments have been addressed

2. Is the manuscript technically sound, and do the data support the conclusions?

Reviewer #2: Yes

3. Has the statistical analysis been performed appropriately and rigorously? 

Reviewer #2: Yes

4. Have the authors made all data underlying the findings in their manuscript fully available?

Reviewer #2: Yes

5. Is the manuscript presented in an intelligible fashion and written in standard English?

Reviewer #2: Yes

6. Review Comments to the Author

Reviewer #2: The authors have carefully addressed all previous comments and concerns, and the revised manuscript now appears suitable for publication. I recommend a final check to ensure that all references cited in the text are properly included in the Literature Cited section, and that the formatting is consistent throughout.

7. PLOS authors have the option to publish the peer review history of their article (what does this mean?). If published, this will include your full peer review and any attached files.

Reviewer #2: **Yes: **DR. Razia Liaqat

---

## [Author Response · Author response to Decision Letter 3]

4 Sep 2025

Response to Reviewers

Reviewer/Editor Comment 1

“I recommend a final check to ensure that all references cited in the text are properly included in the Literature Cited section, and that the formatting is consistent throughout.”

Response:

We thank the reviewer for this important suggestion. We undertook a comprehensive review of the reference list and performed the following steps:

1. Cross-check of in-text citations: Every reference cited in the manuscript text was verified against the References section. All in-text citations are now included in the reference list, and no extraneous references remain.

2. Consistency in formatting: References have been reformatted to comply fully with PLOS ONE requirements (Vancouver style):

• All journal titles are now given in full title case (e.g., Journal of Food Protection rather than J Food Prot.).

• Author initials have been standardized without periods (e.g., Smith AB rather than Smith A.B.).

• Page ranges have been expanded (e.g., S391–S395 rather than S391–S5).

• Organizational reports and government documents (e.g., WHO/FAO, USDA, NOAA, CFR) have been reformatted to include full organizational names, place of publication, publisher, year, URL, and access date.

• Book chapters and monographs (e.g., Thayer DW et al. 1991; Diehl JF. 1995) have been revised to include editors, edition, publisher, location, and page ranges.

• Electronic resources (e.g., USDA FoodData Central) now include full URLs and access dates in the proper format ([cited 2025 Apr 18]).

3. Verification of accuracy: We cross-checked all DOIs, PMIDs, and page numbers to ensure accuracy. We also confirmed that no cited articles had been retracted.

These corrections have been applied consistently across all 46 references. We believe this addresses the reviewer’s concern fully.

---

## [Editor Report · Decision Letter 3]

8 Sep 2025

The effect of gamma irradiation on the stability of vitamin D in select finfish species

PONE-D-24-54557R3

Dear Dr. Smith

We’re pleased to inform you that your manuscript has been judged scientifically suitable for publication and will be formally accepted for publication once it meets all outstanding technical requirements.

Kind regards,

Shafaq Fatima

Academic Editor

PLOS ONE

Additional Editor Comments (optional):

I have checked your response to "Minor Revisions" suggested by the reviewer. As suggested revisions focussed to improving references only, therefore I decided to finalise the decision without fourth review. I strongly realise that authors have waited for so long and responded to all suggested revisions by the reviewers. 

Reviewers' comments:

I as an editor checked the revised version and found that further review is not required. Decision has been made after all references have been cross cheked and improved by the authors. This MS is ready for acceptance.

---

## [Editor Report · Acceptance letter]

PONE-D-24-54557R3

PLOS ONE

Dear Dr. Smith,

I'm pleased to inform you that your manuscript has been deemed suitable for publication in PLOS ONE. Congratulations! Your manuscript is now being handed over to our production team.

Kind regards,

on behalf of

Dr. Shafaq Fatima

Academic Editor

PLOS ONE